# Tumour Genetic Heterogeneity in Relation to Oral Squamous Cell Carcinoma and Anti-Cancer Treatment

**DOI:** 10.3390/ijerph20032392

**Published:** 2023-01-29

**Authors:** Gal Feller, Razia Abdool Gafaar Khammissa, Raoul Ballyram, Mia-Michaela Beetge, Johan Lemmer, Liviu Feller

**Affiliations:** 1Department of Radiation Oncology, University of Witwatersrand, Johannesburg and Charlotte Maxeke Academic Hospital, Johannesburg 2193, South Africa; 2Department of Periodontics and Oral Medicine, School of Oral Health Sciences, Faculty of Health Sciences, University of Pretoria, Pretoria 0084, South Africa; 3Department of Periodontology and Oral Medicine, Sefako Makgatho Health Sciences University, Pretoria 0204, South Africa; 4Retired Professor, Silvela Street, Sandton, Johannesburg 2031, South Africa; 5Retired Professor, Bantry Bay, Cape Town 8005, South Africa

**Keywords:** genetic heterogeneity, genetic diversity, natural selection, cancer-evolution, oral squamous cell carcinoma, driver mutation, passenger mutation, resistance to anti-cancer treatment

## Abstract

Oral squamous cell carcinoma (SCC) represents more than 90% of all oral cancers and is the most frequent SCC of the head and neck region. It may affect any oral mucosal subsite but most frequently the tongue, followed by the floor of the mouth. The use of tobacco and betel nut, either smoked or chewed, and abuse of alcohol are the main risk factors for oral SCC. Oral SCC is characterized by considerable genetic heterogeneity and diversity, which together have a significant impact on the biological behaviour, clinical course, and response to treatment and on the generally poor prognosis of this carcinoma. Characterization of spatial and temporal tumour-specific molecular profiles and of person-specific resource availability and environmental and biological selective pressures could assist in personalizing anti-cancer treatment for individual patients, with the aim of improving treatment outcomes. In this narrative review, we discuss some of the events in cancer evolution and the functional significance of driver-mutations in carcinoma-related genes in general and elaborate on mechanisms mediating resistance to anti-cancer treatment.

## 1. Introduction

Natural selection is a key factor in the evolutionary dynamics of cancer. Positive selection induces an increase in the frequency of cancer-fostering genetic variants in the cell population, thereby promoting tumour progression, while negative selection brings about a reduction in the frequency of cancer-fostering genetic variants, thereby subduing tumour evolution. However, the “branching” of the neoplastic polygenetic tree is not entirely by Darwinian selection but can also be generated by a “mutator phenotype” within tumour cells, giving rise to rates of mutation that can be unfavourable, neutral, or advantageous to the growth of the neoplasm [1].

At the cellular and molecular level, cancer comprises an intricate, complex mosaic of spatially separated subclones of cancer cells, which differ in population size, genetic profile, and phenotypic features. The size of each subclone in the structure of the tumour is determined by the fitness of its cell population (i.e., ability to replicate and survive) and by its genetically conferred selective growth advantage [2,3]. The elaborate interactive communication between cell surfaces of the different subclones influences the biological behaviour of the subclones and has an impact not only on tumour progression but on the choice of and response to anti-cancer treatment [4]. Thus, the intra-tumoural micro- and macro-heterogeneity is the impetus driving the process of cancer evolution and may be associated with failure of anti-cancer treatment [2,3].

Thus, both natural selection and increased rate of random mutations play key roles in cancerization by introducing new genetic variants into the cancer cell population, giving rise to diverse functional phenotypic forms (differences in survival and proliferation rates). These genotypic and phenotypic variations together serve to drive ongoing cancer evolution [5]. However, the temporal evolutionary dynamics enabling the formation of tumour subclones and driving the process of increased clonal fitness, clonal selection, and subsequent clonal expansion are not well understood [6].

The purpose of this narrative review is to consider some of the factors influencing tumour genetic heterogeneity in the context of oral squamous cell carcinoma (SCC) and how this influences response to anti-cancer treatment and hence the prognosis.

## 2. Oral Squamous Cell Carcinoma Genetic Heterogeneity

Oral SCC, like most cancers, is a monoclonal genetic disease arising through accumulation of multiple sequential mutations [7,8,9] that confer upon the mutated keratinocytes and their progeny both fitness advantage and growth dominance over the neighbouring normal cells. This increasing representation of the mutated cells in the affected tissues results ultimately in carcinogenesis [10,11]. However, a field of cancerized oral epithelium may in fact harbour a polyclonal cell population arising from several transformed progenitor/stem cells, each of which has undergone independent clonal expansion with separate clonal divergences, thus multiplying the intra-tumoural micro- and macro- heterogeneity [12,13] (Figure 1).

The incidence, prevalence, and other epidemiological features of oral SCC differ greatly among populations living in different geographic locations and among various ethnic groups within the same population, probably owing to environment-specific factors, to ethnic-specific high-risk factors, and to genetic predisposition [14,15]. Nevertheless, oral SCC always arises within the field of pre-cancerized oral epithelium either *de novo* or from pre-existing potentially malignant lesions such as non-homogeneous leukoplakia, erythroplakia, or submucous fibrosis [16] (Figure 1).

Surgical resection is the preferred first line of treatment, with adjunctive treatment (radiotherapy with or without chemotherapy) being added in cases of advanced disease or in cases deemed to be at significant risk of recurrence [17,18]. The overall five-year survival rate of persons with oral SCC ranges from 55% to 70% [17,19,20]. The most critical prognostic factor is the stage of advancement of the disease at the time of diagnosis [16], and an important cause of treatment failure and low rate of survival is the great degree of genetic heterogeneity of oral SCC [21,22].

In addition to being molecularly heterogeneous, oral SCC also displays heterogeneity with regard to the diversity of its cell population, which includes tissue-specific cancer stem/progenitor cells, transit-amplifying cancer cells, post-mitotic cancer cells at different stages of maturation, and de-differentiated cancer cells. All of these contribute to the genetic and phenotypic heterogeneity of the carcinoma [10].

## 3. Cancer Driver Genes

Cancer evolution results in the development of intra-tumoural micro- and macro-genetic heterogeneity, both of which play roles in resistance to chemotherapy, thus contributing to failure of treatment. While genetic micro-heterogeneity is generated by and refers to the molecular aspects and dynamics of clonal evolution, macro-heterogeneity comes about by the evolution of subclones, which subsequently expand along the complex branch trajectories of an evolutionary polygenetic tree [7].

Genomic instability and mutated phenotypes, on the one hand, power the generation of many new mutations during clonal evolution, providing the impetus towards tumour micro-heterogeneity. Macro-heterogeneity with its subclonal diversity, on the other hand, provides the substrate upon which natural selection drives the complex adaptive process of cancer evolution through expansion of subclones harbouring advantageous mutations that confer upon the subclones the benefits of cellular fitness and survival [7] (Figure 1).

Cancer driver genes are mutant genes that confer the advantage of selective fitness upon cancer cells [23], and the mutations that generate cancer driver genes are termed driver mutations [24]. The mutated cancer driver genes can then be classified either as oncogenes or as tumour suppressor genes (anti-oncogenes) [25,26]. Activating mutations in proto-oncogenes give rise to oncogenes that upregulate proliferation of cancer cells, leading to their uncontrolled proliferation, while inactivating mutations in tumour suppressor genes (anti-oncogenes) result in downregulation of processes that control cell cycle progression, thereby enabling propagation of altered DNA to daughter cells, impeding apoptosis with consequent prolonged cell survival [11,16].

Genomic instability may be brought about by loss-of-function mutations in *mlh1* and *msh2* genes, which are involved in repair of mismatch DNA bases that have been incorporated during DNA replication, and in nucleotide excision repair pathways that correct covalent alterations to DNA induced by chemical mutagens; by chromosomal aberrations including aneuploidy, deletions, translocations, and amplifications; by dysregulating activity of checkpoints that control cell-cycle progression; and by upregulation of functional activity of oncogenes conferring cellular autonomy in proliferative signals or downregulation of functional activity of tumour suppressor genes, resulting in reduced anti-oncogenic activity [11,12,27,28,29]. Although genomic instability is an early event in the carcinogenic process, the oncogenes and tumour suppressor genes that play a role in creating genomic instability in normal keratinocytes, thus driving the process of initial transformation, also contribute to the progression of precancerous keratinocytes to cancer cells and later to tumour progression [12] (Figure 1).

In this regard, loss-of-function mutation in the TP53-tumour suppressor gene, which is prevalent in oral SCC, promotes tumour genetic microheterogeneity since it allows replication of damaged DNA and proliferation of mutant keratinocytes, having among themselves an immense diversity of molecular profiles [22]. As reported by the Indian Project Team of the International Cancer Genome consortium [30], the mutational landscape of oral SCC (gingivo-buccal sites) is characterized by frequently mutated specific cancer-relevant genes such as USP9X, MLL4, ARID2, UNCBC, and TRPM3, while other frequently mutated cancer-relevant genes such as TP53, FAT1, CASP8, HRAS, and NOTCH1 are common to all head and neck SCCs [30,31]; and a study that profiled cancer-related gene mutations in oral SCC of Japanese patients revealed that the most frequent mutations were TP53 (62%), NOTCH 1 (26%), and CDKN2A (19%) [32].

It is not always possible to differentiate between driver genes distinct to either macro- or micro-heterogeneity. This is because some driver mutations are common to the evolution of both the original malignant clone and to the subsequent subclones during the phase of tumour progression. Further, as there are differences in the clinical and histopathological features of and survival rates associated with SCC at different oral subsites [33], some researchers are of the opinion that oral SCC at various subsites should be considered as distinct, separate disease entities [34,35,36]. However, the underlying molecular mechanisms (oncogenes, anti-oncogenes, genetic heterogeneity features, etc.) that account for the different bio-clinical behaviour of SCC at the various oral subsites are undetermined.

In any event, it appears that the few prominent driver genes that are most frequently mutated in cancers; e.g., TP53 are not in themselves sufficient to promote the complex process of carcinogenesis; and recurrent but infrequently mutated “longtail” genes, many of which have not as yet been characterized, can promote oncogenesis. These longtail, potentially cancer-relevant genes have singularly little biological and clinical significance, but when several such singular genes interact, they have the synergistic capacity to promote cancerization [31,37].

In about 30% of cancers, the process of cancerization appears to be caused by recurrent driver gene mutations in tissue-specific sets of genes clustered into distinct modules and in the remainder, by non-specific and variable driver mutations in diffuse sets of potentially cancer-relevant genes, which cooperatively promote cancerization [38].

## 4. Driver and Passenger Mutations

Of the many thousands of genomic mutations generated by cancer cells, only a very small number, termed driver mutations (drivers), are essential for cancer initiation and progression. Drivers confer upon cancer cells an increased fitness and proliferative advantage, resulting in their capacity for uncontrolled proliferation and prolonged survival, with subsequent disruption of homeostasis. In contrast, the vast majority of genomic mutations observed in cancer cells, which are termed passenger mutations (passengers), do not directly promote cancerization and have but little biological significance in the carcinogenic process [7,9,39,40,41].

Most passengers are phenotypically neutral, but some may be either moderately or highly deleterious to the cancer itself. The highly deleterious passenger mutations usually become extinct by negative natural selection, but those passengers with a neutral or moderately deleterious phenotype may evade natural selection and accumulate. Collectively, a high burden of deleterious passengers has the potential to cause damage to the cancer through several mechanisms, including by the generation of proteotoxic stress, by inducing anti-tumour immune responses, and by prompting dysfunctional cellular activity. This may influence the overall biological behaviour of the cancer and, among others, be a factor in determining tumour dormancy, tumour growth rate, spontaneous regression, and response to cancer chemotherapy [40,42]. In this regard, it has been proposed that there is another category of passengers, namely “mildly beneficial” passengers or “mini drivers”, which have the capacity to confer some slight fitness advantage to the mutated cells, thereby to some extent counteracting the effects of deleterious passengers [43].

It appears that chemotherapy to increase deleterious passenger load, which in turn may induce chromosomal instability, aneuploidy, and high levels of DNA damage with subsequent cancer cell death, may bring about beneficial clinical outcomes. Thus, increasing passenger load by chemotherapy may outperform driver-targeted therapies, possibly because elimination of specific drivers may promote the evolution of new drivers with increased fitness-advantage properties, which then confer tumour resistance to treatment [40].

## 5. Tumour Genetic Heterogeneity and Phenotypic Plasticity in Relation to Tumour Microenvironment

Physiologically, cellular function and structure are maintained by systems or circuits of regulatory genes organized in networks. Changes in the profiles of such gene circuits may be induced by cytogenetic mutations, by epigenetic modifications, and/or by non-genetic micro-environmental factors including chronic inflammation and altered mechanical properties of extracellular matrix. Such genetic changes have an impact on cell attachment, proliferation, differentiation and apoptosis, and consequently on initiation, promotion, and progression of cancer [44,45].

As mentioned above, tumour phenotype is determined by tumour genotype and its gene regulatory circuits, by epigenetic modification, and by factors in the microenvironment of the tumour [46]; and tumour genetic heterogeneity is the force driving the rapid adaptation of cancer clones to the pressures imposed upon them by changing microenvironmental circumstances, by contextual immune responses, and by anti-cancer treatment [41].

Changes in the microenvironmental ecosystem of a tumour affect the nature of the intra-tumoural cellular, genetic, epigenetic, and phenotypic heterogeneity [47], accounting for the significant variations in genetic makeup, histopathological features, clinical course, and response to treatment of carcinomas affecting the same tissues in the same anatomical sites but in different persons [41]. There is also significant inter-tumoural genetic heterogeneity among different tumours of the same type affecting the same anatomical site and with similar phenotypical features and histopathological morphology [46]. In addition, there are spatial, genotypical, phenotypical, and morphological differences within a particular carcinoma [46], probably brought about by its intra-tumoural genetic heterogeneity and by changing micro-environmental conditions such as the biophysical properties, oxygen concentration, growth factors, and cytokine concentrations in the extracellular matrix [4,41].

The great variability of inter-tumour and inter-person genetic heterogeneity is probably owing to the uniqueness of each person’s genomic/genetic instability; to driver, passenger, and longtail mutations; and to patient-specific selective pressures stemming from the person’s endocrinological, immunological and nutritional status, general health, life style, and past exposures to therapy. Consequently, each tumour-specific genetic and phenotypic combination is not only unique, but it also changes over time, as also does the tumour’s response to anti-cancer treatment [4].

Tumour clonal/subclonal neoantigens (new proteins that may develop on cancer cells following mutations in tumour DNA) can trigger diverse immunoinflammatory reactions within the tumour microenvironment, some of which may be tumour-suppressive and others tumour-promotive. Neoantigen-specific anti-cancer immune responses, either natural or brought about by neoantigen vaccines, can impose subclone-specific negative selective pressures, thus contributing to clonal evolution and to tumour subclonal composition [48].

Cancer is a complex ecosystem of competing clones; its tissue microenvironment is multifactorially adaptive, incorporating cancer cells, non-cancer cells, and growth factors [49,50]. The dysregulated intrinsic genetic circuits and signalling pathways, which mediate prolonged survival and increased proliferation of cancer cells, can also induce the production of immune and inflammatory mediators. Consequently, the cancer-associated inflammatory microenvironment has the capacity to further increase proliferation and survival of cancer cells and to promote angiogenesis and evasion of anti-cancer immune responses [51].

Microenvironmental stressors such as hypoxia and inflammation, as mentioned above, may induce epigenetic changes that in turn influence expression and function of gene regulatory circuits. This enables metabolic reprogramming of and confers improved fitness upon the transformed/cancer cells, resulting in clonal adaptation and expansion. Thus, epigenetically driven, non-mutational reorganization of gene regulatory circuits promotes tumour genetic heterogeneity and phenotypic plasticity [45].

A tumour microenvironment is characterized by dysregulated reciprocal communication between cells and between cells and the extracellular matrix, by altered cell metabolism and phenotype, and by dysregulated vasculature and oxygenation [45,52]. The fibroblasts of the carcinoma-associated stroma are functionally impaired by the same genotoxic or mutagenic agents/events that mediated the transformation of the carcinomatous cells [51,52] and later by dysregulated paracrine, autocrine, and direct cell-to-cell signalling as well as by microenvironmental changes [45,51,52]. Thus, the intricately dysregulated reciprocal interactions between the various microenvironmental elements promote genetic heterogeneity of mutated cells and influence cancer evolution, cancer phenotype, and response to radiotherapy and chemotherapy. In addition to directly targeting and killing cancer cells, anti-cancer therapy should also aim at altering cancer-supporting microenvironmental elements by reducing the vasculature to promote local hypoxia and by reducing local inflammation [49].

## 6. Intratumour Heterogeneity and Resistance to Cancer Chemotherapy

As the evolutionary dynamics of genetically heterogeneous tumours are predominantly mediated by competition between subclones with different genetic makeup and fitness, it would be advantageous to determine the dose, time, and sequence of anti-cancer chemotherapy according to the genetic profile and evolutionary dynamics of the tumour when treatment is started [53]. It would also be clinically beneficial to be able to evaluate the genetic makeup of the tumour before, during, and after chemotherapy because this information might allow personalization of the drug therapy employed and reduce the risk of drug resistance. Unfortunately, because of the technical challenges and prohibitive cost, this is not feasible at present [54].

In this regard, adaptive adjustment of the dose, sequence, and period of delivery of the anti-cancer agents (“adaptive therapy”), may promote beneficial anti-cancer competition between drug-sensitive and drug-resistant clones, thus preventing selection and subsequent expansion of drug-resistant clones and delaying treatment failure [53].

Cytotoxic chemotherapy-induced genotoxicity and mutagenesis bring about an increase in the intra-tumoural mutational burden, genetic heterogeneity, and selective pressures, all of which facilitate the evolution of drug-resistant clones, thus causing failure of treatment and relapse [13,49,55]. After a few rounds of cytotoxic chemotherapy, most of the drug-sensitive clones within the tumour are either eliminated or substantially reduced, consequently providing a more favourable environment for the expansion of the surviving drug-resistance subclones because they are then released from the previous suppressive competition with drug-sensitive subclones for metabolic resources and space. This “competitive release” principle of co-evolutionary ecology and population dynamics appears to be the main mechanism by which genetically heterogeneous cancers develop resistance to chemotherapeutic agents [49,56,57].

In targeted therapy, a target-drug attacks and neutralizes a specific intracellular molecular pathway that is essential for prolonged survival or for increased proliferation of cancer cells [41]. Resistance to molecularly targeted monotherapy is caused either by acquisition of additional mutations that enable bypassing the intracellular molecular signalling pathway attacked by the target drug, thus maintaining the cancer cell oncogenic activity [49], or, again, by competitive release, as explained above. The target therapy-induced diminution in the size of the target clone gives a competitive growth advantage to pre-existing clones that do not carry the targeted proteins, resulting in expansion of tumour-resistant clones, causing treatment failure and relapse [55].

The principles of the competitive release phenomenon can also be exploited for anti-cancer treatment. By maintaining residual drug-sensitive subclones within the tumour cell population upon withdrawal of chemotherapy, these surviving subclones may be capable of competing with and suppressing the growth of drug-resistant clones within the tumour [58]. Thus, treating genetic heterogeneous cancers by employing evolutionary and ecological principles aiming at maintaining competing subclones that mutually suppress each other’s growth may limit cancer progression and reduce frequency of relapse [6,56].

## 7. Radiation Therapy in Relation to Tumour Heterogeneity

In vitro studies show that, as is the case with drug-induced resistance to chemotherapy, radiotherapy with conventional fractionated ionizing radiation (IR) usually induces selection of IR-resistant subclones of cells that show increased potential for damaged-DNA repair and prolonged survival. Consequently, the surviving subclones have the capacity both to prevail against the anti-proliferative effects of subsequent fractionated schedules of IR and to intensify the oncogenic potential of the irradiated subclones [59].

IR-mediated genetic heterogeneity may induce cancer cells to acquire altered activity of a great number of intracellular signalling pathways that promote radiation resistance. These include downregulation of inhibitory damaged-DNA repair and of cell-death pathways and upregulation of damaged-DNA repair and of anti-apoptotic pathways. In addition, pathways associated with stemness and acquisition of a cancer stem cell phenotype as well as with reprogramming of cellular metabolic activities and with increased production of inflammatory mediators supporting cell proliferation are also upregulated [15,52,60]. In turn, this de novo IR-mediated oncogenesis and the existing pre-treatment tumour genetic heterogeneity may together bring about failure of treatment and account for the poor prognosis of oral SCC treated with radiotherapy [61].

The progressive evolution of adapting IR-resistant subclones in response to the significant selective pressure exerted by multiple sequential rounds of fractionated IR may frequently leads to treatment failure and relapse [59,62]. It appears that in comparison to conventional fractionation, hypofractionated radiotherapy in which the total dose of the IR delivered is divided into fewer but larger (>2 Gy) dose fractions is associated with better clinical outcome but is also associated with an increased risk of late IR-induced toxicity [62].

In order to improve IR-induced cancer response to and the efficacy of radiotherapy, it is important to know the spatial and temporal distribution of radioresistance/radiosensitivity factors within the genetically heterogeneous irradiated tumour and within its microenvironment because this may facilitate the selection of the most favourable personalized fractionation protocol for best clinical outcomes [63].

## 8. Conclusions

Carcinogenesis is a dynamic, complex adaptive process driven by genetic mutations, clonal selection and expansion, and genetic, epigenetic, and microenvironmental heterogeneity within a tumour-specific ecosystem. The inherent genetic heterogeneity, phenotypic plasticity, and adaptability of advanced cancers are critical determinants of the response to anti-cancer treatment.

The critical role that clonal evolution plays in cancerization, targeting only the most dominant subclone but disregarding other genetic and non-genetic components of the tumour, is a major cause of treatment failure. Therefore, an effective personalized anti-cancer treatment strategy should incorporate and integrate elements that are based on selective evolutionary principles; that target the dynamic adaptive genotypical, phenotypical, and epigenetic determinants of cancerization; and that disturb microenvironmental factors of tumour growth.

## Figures and Tables

**Figure 1 ijerph-20-02392-f001:**
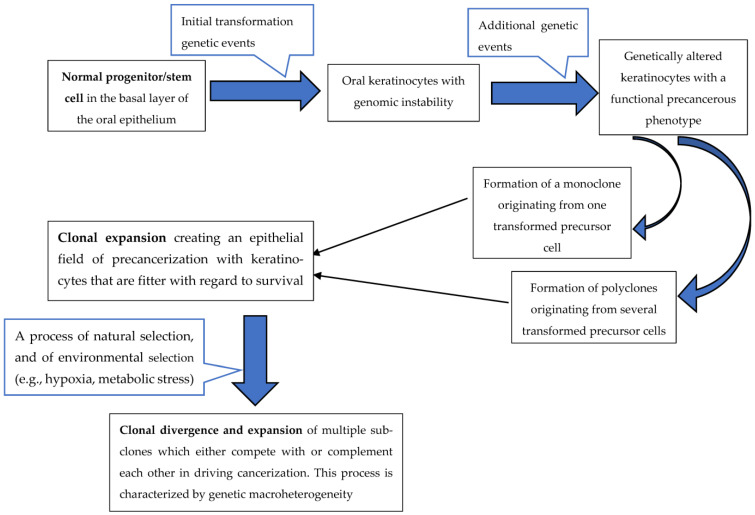
Schematic presentation of the cancerization process adapted from Feller et al., 2013 (12).

## Data Availability

Not applicable.

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
