# Peer review of "Tumour Genetic Heterogeneity in Relation to Oral Squamous Cell Carcinoma and Anti-Cancer Treatment"

_ijerph, 2023, doi:10.3390/ijerph20032392_

Round 1
Reviewer 1 Report
It is a very interesting manuscript, it applies theories of ecology and evolution to the carcinogenesis process, it addresses the concept of genetic heterogeneity in a simple and understandable way. My comments to improve this manuscript would be:
1. In the "Cancer Driver genes" section, he touches on a controversial point regarding the fact that oncogenes and tumor suppressor genes are particularly linked to macro-heterogeneity, which is related to the development of sub-clones, however, there is evidence where these oncogenes or tumor suppressor genes could also be responsible for clonal evolution and therefore also be related to micro-heterogeneity, perhaps examples of which genes may be linked to macro or micro-heterogeneity would help to better differentiate the application.
2. As the central gene of the "Cancer Driver genes" theme, only TP53 is mentioned, however, other genes should be mentioned, such as those that participate in the genomic stability process as "caretaker"
3. Regarding the percentage of most frequent mutations in the Japanese population, it would be good to include the figure corresponding to CDKN2A (page 3, line 123)
4. The document has many extra spaces, check the format (page 2, line 59; page 3, line 104, 134, etc).
5. In the references section, complete references 16 and 33.
Author Response
Dear Editor,
We want to thank you the Editor and the Reviewers for their hard work reviewing our paper, and for their constructive suggestions.
RESPONSE TO REVIEWER 1
1.and 2.
The section ‘Cancer driver genes’ has been revised. We have deleted the last paragraph of the section on page 9, and we have introduced a new paragraph on page 3, fourth paragraph that reads:
‘Genomic instability may be brought about by loss-of-function mutations in mlh1 and msh2 genes which are involved in repair of mismatch DNA bases that have been incorporated during DNA replication, and in nucleotide excision repair pathways that correct covalent alterations to DNA induced by chemical mutagens; by chromosomal aberrations including aneuploidy, deletions, translocations and amplifications; by dysregulating activity of checkpoints that control cell-cycle progression; and by upregulation of functional activity of oncogenes conferring cellular autonomy in proliferative signals, or downregulation of functional activity of tumour suppressor genes resulting in reduced anti-oncogenic activity (11, 12, 27-29). Although genomic instability is an early event in the carcinogenic process, the oncogenes and tumour suppressor genes that play a role in creating genomic instability in normal keratinocytes thus driving the process of initial transformation, also contribute to the progression of precancerous keratinocytes to cancer cells, and later to tumour progression (12) (Figure 1).’
- Please also see the new paragraph added under the heading ‘Cancer driver genes’, 6th paragraph that reads:
‘It is not always possible to differentiate between driver genes distinct to either macro- or micro-heterogeneity. This is because some driver mutations are common to the evolution of both the original malignant clone, and to the subsequent subclones during the phase of tumour progression. Further, as there are differences in the clinical and histopathological features of, and survival rates associated with SCC at different oral subsites (33), some researchers are of the opinion that oral SCC at various subsites should be considered as distinct, separate disease entities (34-36). However, the underlying molecular mechanisms (oncogenes, anti-oncogenes, genetic heterogeneity features, etc) that account for the different bio-clinical behaviour of SCC at the various oral subsites are undetermined.’
Thus, as we have explained in the newly revised section, it is not always possible to differentiate between driver genes distinct to either micro- or microheterogeneity. This is because some driver mutations are common to the evolution of both the original malignant clone, and to the subsequent subclones during the phase of tumour progression. Therefore, we have opted not to categorically differentiate between micro- and microheterogeneity.
- We have added the missing frequency (19%). Thank you.
- We have fixed the reference list, and the typographical errors.
Reviewer 2 Report
The topic covered is very interesting and well presented.The weakness of the article is the absence of images and tables that would be useful to readers. Specifically, I would introduce a table and an image describing the driver and passenger genes that have a main role in the heterogeneity of the OSCC and a summary image of the tailoring therapy that can result.
Author Response
Dear Editor,
We want to thank you the Editor and the Reviewers for their hard work reviewing our paper, and for their constructive suggestions.
RESPONSE TO REVIEWER 2
- Please see the newly introduced Figure 1 that illustrates microheterogeneity.
- With regard to generating a Table/image ‘describing the driver and passenger genes that have a main role in the heterogeneity of OSCC’, we have explained in the newly revised text why this may be problematic. Please see the fourth paragraph under the heading ‘Cancer driver genes’ on page 3.
- As the molecular alterations of oral SCC vary among population groups in different geographic locations and is greatly influenced by ethnic and genetic factors, by environmental exposures, and individual life-style practices, we are of the opinion that composing a Table with definitive molecular alterations applicable for all populations groups is not practical. Nevertheless, the issues are discussed in the text and touched upon in Figure 1.
- The purpose of this article was to consider some factors influencing tumour genetic heterogeneity in the context of oral SCC, and how these factors influence anti-cancer treatment decisions. However, none of the co-authors of this article is either a qualified clinical oncologist nor has any clinical experience with regard to anti-cancer targeted therapy. Thereby, in order to avoid any possibility of misleading the reader, we have decided not to construct, as suggested, an image illustrating verified clinically available molecular targets that are currently in practical use.
- Please see the new paragraphs added under the heading ‘Cancer driver genes’, 4th and 6th paragraph that reads:
New fourth paragraph
‘Genomic instability may be brought about by loss-of-function mutations in mlh1 and msh2 genes which are involved in repair of mismatch DNA bases that have been incorporated during DNA replication, and in nucleotide excision repair pathways that correct covalent alterations to DNA induced by chemical mutagens; by chromosomal aberrations including aneuploidy, deletions, translocations and amplifications; by dysregulating activity of checkpoints that control cell-cycle progression; and by upregulation of functional activity of oncogenes conferring cellular autonomy in proliferative signals, or downregulation of functional activity of tumour suppressor genes resulting in reduced anti-oncogenic activity (11, 12, 27-29). Although genomic instability is an early event in the carcinogenic process, the oncogenes and tumour suppressor genes that play a role in creating genomic instability in normal keratinocytes thus driving the process of initial transformation, also contribute to the progression of precancerous keratinocytes to cancer cells, and later to tumour progression (12) (Figure 1).’
New sixth paragraph
‘It is not always possible to differentiate between driver genes distinct to either macro- or micro-heterogeneity. This is because some driver mutations are common to the evolution of both the original malignant clone, and to the subsequent subclones during the phase of tumour progression. Further, as there are differences in the clinical and histopathological features of, and survival rates associated with SCC at different oral subsites (33), some researchers are of the opinion that oral SCC at various subsites should be considered as distinct, separate disease entities (34-36). However, the underlying molecular mechanisms (oncogenes, anti-oncogenes, genetic heterogeneity features, etc) that account for the different bio-clinical behaviour of SCC at the various oral subsites are undetermined.’
We want to once again thank the reviewers.
Yours Sincerely,
Razia Khammissa
Associate Professor and Head of Department: Periodontics and Oral Medicine
School of Dentistry, University of Pretoria, South Africa
24 January 2023